# Updated Checklist of Chondrichthyan Species in Croatia (Central Mediterranean Sea)

**DOI:** 10.3390/biology12070952

**Published:** 2023-07-03

**Authors:** Pia F. Balàka, Pero Ugarković, Julia Türtscher, Jürgen Kriwet, Simone Niedermüller, Patrik Krstinić, Patrick L. Jambura

**Affiliations:** 1Department of Palaeontology, University of Vienna, Josef-Holaubek-Platz 2 (UZA II), 1090 Vienna, Austria; tuertscher.julia@gmail.com (J.T.); juergen.kriwet@univie.ac.at (J.K.); 2World Wide Fund for Nature Adria (WWF Adria), Gundulićeva 63, 10 000 Zagreb, Croatiapkrstinic@wwfadria.org (P.K.); 3Vienna Doctoral School of Ecology and Evolution (VDSEE), University of Vienna, Djerassiplatz 1, 1030 Vienna, Austria; 4World Wide Fund for Nature Mediterranean Marine Initiative (WWF MMI), Via Po 25/c, 00161 Rome, Italy; simone.niedermueller@wwf.at

**Keywords:** cartilaginous fishes, endangered species, Adriatic Sea, Red List (IUCN), citizen science, MECO project

## Abstract

**Simple Summary:**

Relatively little is known about cartilaginous fishes (sharks, rays, skates, and chimaeras) in the Adriatic Sea. Still, many tons of these endangered animals are caught every year and not many protective measures are in place. How this exploitation affects their populations today, in the past and in the future is unknown. Our study outlines a list of all species known to occur in Croatia based on historical as well as new data. This updated information establishes a baseline for further research and can help policymakers make informed decisions on conservation measures for these fishes.

**Abstract:**

Almost half of all chondrichthyan species in the Mediterranean Sea are threatened with extinction, according to the IUCN Red List. Due to a substantial lack of access to data on chondrichthyan catches in the Mediterranean Sea, especially of threatened species, the implementation of conservation measures is extremely insufficient. This also concerns the Adriatic Sea. Here we present a detailed and up-to-date assessment of the species occurring in Croatian waters, as the last checklist of chondrichthyans in Croatian waters was conducted in 2009. Occurrence records from historical data, literature and citizen science information have been compiled in order to present a comprehensive list of species occurrences. We found 54 chondrichthyan species between 1822 and 2022, consisting of a single chimaera, 23 rays and skates, and 30 shark species. Here, four additional species are listed but are considered doubtful. Five species are reported here for the first time for Croatian waters that were not listed in the survey from 2009. Nearly one-third of the species reported here are critically endangered in the entire Mediterranean Sea, based on the IUCN Red List. Additionally, we revisited the Croatian records of the sandtiger shark *Carcharias taurus* Rafinesque, 1810 and discussed its potential confusion with the smalltooth sandtiger shark *Odontaspis ferox* (Risso, 1810). Our results thus provide novel insights into the historical and current distribution patterns of chondrichthyan fishes in the Croatian Sea and provide a basis for further research as well as conservation measures.

## 1. Introduction

Chondrichthyes (chimaeras, sharks, rays, and skates), or cartilaginous fishes, are an evolutionarily successful group of vertebrates that originated as early as the Late Ordovician, around 450 million years ago [1]. Due to their long evolutionary history, they are intimately intertwined with their marine ecosystems. Consequently, they are crucial for ecosystem function and maintenance [2]. The modern Mediterranean Sea is approximately 5.3 million years old and was formed during the Zanclean flooding event after the Messinian Salinity Crisis [3]. With 86 chondrichthyan species, the Mediterranean Sea is regarded as a biodiversity hotspot for this group [4]. Historically, however, cartilaginous fishes have been caught and consumed by humans in this area for thousands of years [5].

Today, the Mediterranean Sea is among the most overexploited marine areas in the world, with 62% of fish stocks being depleted to biologically unsustainable levels [6]. Once populations are depleted, they need a long time to recover due to the fact that chondrichthyans are K-strategists that only bear a few offspring and take a long time to mature [7]. Due to this particular vulnerability, almost half of the shark species in the Mediterranean are currently threatened with extinction [8]. Bycatch is considered the main threat for sharks and batoids (rays and skates) in the Mediterranean Sea, where they are caught unintentionally by small- and large-scale trawls as well as net multispecies fisheries [8]. Since certain species are protected, fishermen are obliged to release them back into the sea, ideally when they are still alive [9]. However, there is a large variability in survival rates after release amongst these fishes, depending on the species [9]. Sharks and batoids are also caught intentionally to compensate for declining catches of bony fish and are sold as local specialties at fish markets, where they are often skinned and labelled with misleading names, thus confusing the consumer [10]. Furthermore, endangered species are often targeted by recreational fishermen as trophies [11]. An unknown extent of illegal fishing further contributes to the overall high mortality of sharks and batoids [8]. All these factors contribute to the strong decline of chondrichthyan populations in the Mediterranean.

Not only are fish stocks heavily exploited, but there is also scarce data available on the quantity of animals caught and the species composition of catches. Also, the kinds of fisheries involved, and the fishing gear used are rarely reported. Thus, the population statuses of many elasmobranch species (sharks, rays, and skates) in Europe cannot be assessed by the IUCN due to a severe lack of data, resulting in their statuses being data deficient (DD) [12].

The EU and the Croatian Ministry of Agriculture collect and publish only limited data. The Croatian State Bureau of Statistics reports a total of 61,574 tons of catches for all fish, crustaceans, shellfish and molluscs for 2021 [13]. The FAO similarly reports 62,422 tons in live weight of fish and fishery products from Croatia in 2021 [14]. Of this, according to Eurostat, a total of 3173 tons of live sharks, batoids, and chimaeras were caught in the Mediterranean and the Black Sea in 2021, with Croatia contributing 282 tons [15]. Which species were caught is not specified. Only the Data Collection Framework (DCF) by the European Commission specifies some species, but the Croatian report from 2021 included only 23 species of chondrichthyans, 15 of which had no registered landings this year [16]. Consequently, the real number of sharks, rays, skates, and chimaeras caught in Croatia and the species composition of the catches remain difficult to assess.

In order to accomplish the implementation of efficient protective measures and conservation legislation for chondrichthyans in Croatian waters, a faunal checklist is an important baseline for identifying the occurrence and number of endangered species in this area. The last checklist for Croatia was done in 2009 and found 52 chondrichthyan species (one chimaeroid, 23 batoid, and 28 shark species) to be present in Croatian waters [17]. The objective of the current study is to revise and update this checklist based on data collected from museum inventories, published information and citizen science observations. The updated checklist revisits and discusses historical and current distribution patterns and is intended to serve as an integral basis for future population assessments and management efforts for chondrichthyan fishes in Croatia’s Exclusive Economic Zone.

## 2. Materials and Methods

The Exclusive Economic Zone (EEZ) of Croatia has been used as a boundary for Croatian waters [18]. It covers an area of 55,961 km^2^, which represents about 40% of the Adriatic Sea [19]. Records from the neighbouring countries of Slovenia, Montenegro and Bosnia and Herzegovina have not been included in the data set presented here.

Occurrence data for chondrichthyan fishes in the Croatian EEZ were collected until February 2022. Data came from three main sources: (1) published literature; (2) historical data (museum inventories); and (3) citizen science. The data collected ranges from the 19th to the 21st century.

An extensive literature search was done using bibliographic databases such as Google Scholar and shark-references.com [20]. Key words such as “sharks”, “rays”, “skates”, “chimaeras”, “Chondrichthyes”, “Elasmobranchii”, “Adriatic”, “Croatia”, and “Mediterranean” in English and Croatian were used for data mining. References for each record are provided in Appendix A. All published records were reviewed and verified.

Historical information comes from chondrichthyans which are housed in museum collections. Information about these specimens was collected from papers, public collection catalogues, or provided by corresponding museum staff. All museums located in Croatia suffered severe damage during the Second World War (1939–1945) and the Croatian War of Independence (1991–1995). The Dubrovnik Natural History Museum was also damaged during an earthquake in 1979. Many specimens were lost as a result of these events. Entries from collection catalogues were nonetheless included (note that no new species were added this way). Pictures of specimens (e.g., Figure 1B) from the Natural History Museum Vienna were taken using an Olympus OM-D E-M1 with a mounted Olympus M.Zuiko 12–50 mm lens.

Citizen science records were provided by the MECO Project [21], where data was collected via social media by means of data mining of social networks and groups that deal with recreational and professional fisheries and through personal contacts. Only sightings with video or photographic evidence were considered and used for identification.

Each record was identified at the lowest possible rank (i.e., down to the species level). Nomenclature and systematic classification follow Eschmeyer’s Catalog of Fishes [22] and the World Register of Marine Species (WoRMS) [23]. Additional information, i.e., location, sex, body size, weight, depth, and used fishing gear, was collected where possible. The ontogenetic stage of individuals was estimated based on body size data when applicable (see Appendix A).

A heatmap of all chondrichthyan species in Croatia was created. Spatial information was obtained from museum records, literature, and citizen science records. Each record was put in QGIS version 3.22.3-Białowieża and heatmaps were created. Reports of more than one specimen from the same source were treated as a single data point, which is especially important for all records from Kirincić and Lepetić (1955) [24], as their work contributed 429 records for Dubrovnik alone. An approximate location was taken from Google Maps when the Global Positioning System (GPS) position was not provided. The satellite map by Esri [25] was used as a base map. Additionally, a shapefile of the European coastline was taken from the European Environment Agency [26]. A shapefile of Croatia was taken from DIVA-GIS [27], and a shapefile of Croatia’s Exclusive Economic Zone was taken from Marineregions.org [28].

## 3. Results

### 3.1. Systematic Analysis

This checklist consists of 999 records for at least 2628 individuals. Around 176 records (18%) are from historical data, and 114 records (11%) are from citizen science data. Around 709 records (71%) are derived from published data. However, 429 of these records are from Kirincić and Lepetić (1955) alone [24]. The oldest record is from 1822 and the latest is from 2022; thus, data from 200 years of chondrichthyan reports are included here.

The systematic collection of historical, literature, and citizen science data added up to records of 54 chondrichthyan (one chimaeroid, 23 batoid, and 30 shark) species that have been identified to occur in the Croatian Sea. This amounts to 63% of all 86 chondrichthyan species recorded in the Mediterranean Sea [4]. The presence of four additional species, namely *Raja undulata* Lacepède, 1802, *Rhinobatos rhinobatos* (Linnaeus, 1758), *Carcharias taurus* Rafinesque, 1810, and *Sphyrna tudes* (Valenciennes, 1822), is here considered ambiguous (see extended discussion) and remains to be confirmed in the future.

### 3.2. Analysis of IUCN Red List Categories and Citizen Science Data

Thirty-seven (69%) of all 54 chondrichthyan species in Croatian waters identified during this study are globally in a threatened state (either critically endangered, endangered or vulnerable), according to the IUCN [12]. Thirty-one species (57%), i.e., more than half of all species found in Croatian waters, are considered threatened in the Mediterranean Sea [12]. Eight (15%) species out of 54 in Croatian waters are critically endangered globally, according to the IUCN (Figure 2A) [12]. Sixteen (30%) out of 54 species in Croatian waters are considered to be critically endangered in the Mediterranean Sea by the IUCN (Figure 2B) [12]. Twelve (22%) out of 54 species are endangered globally and nine (17%) in the Mediterranean Sea (Figure 2A,B) [12]. Seventeen (32%) out of 54 species in Croatian waters are considered vulnerable globally (Figure 2A) and six (11%) in the Mediterranean Sea (Figure 2B) [12]. The global IUCN status of two species found in Croatian waters is data deficient (Figure 2A), and nine species are data deficient in the Mediterranean Sea (Figure 2B) (Table 1) [12].

Citizen science records provided by scuba divers, snorkelers, beachgoers and sightings from boats encountered 16 (30%) out of all 54 species in Croatian waters (Figure 2C). On the other hand, professional and recreational fishers that provided data for citizen science encountered 21 (39%) out of all 54 species (Figure 2D). The fishing techniques used by these fishermen include angling, longlining, gill nets, beach seine and trawling (see Appendix A). In total, citizen science data alone reported 24 (44%) out of all 54 species in Croatian waters.

## 4. Discussion

### 4.1. Comparison to Previous Checklists

In 2009, Alen Soldo and Nedo Vrgoč compiled a checklist of cartilaginous fishes in Croatia [17]. In our updated list, five additional species were found to be present in Croatian waters: *Dipturus nidarosiensis* (Storm, 1881), *Raja montagui* Fowler, 1910, *Squatina aculeata* Cuvier, 1829, *Alopias superciliosus* (Lowe, 1841), and *Scyliorhinus duhamelii* (Garman, 1913).

In 2020, Kovačić et al. provided a checklist of fishes in the Adriatic Sea, including 55 chondrichthyan species [39]. The following species were reported for the Adriatic Sea which we could not confirm in our evaluation here: *Hexanchus nakamurai* Teng, 1962, *Carcharias taurus*, *Rhizoprionodon acutus* (Rüppell, 1837), and *Rhinoptera marginata* (Geoffroy St. Hilaire, 1817) [39]. Our new checklist includes two hitherto unrecognized species that were not listed in the checklist of Adriatic fishes provided by Kovačić et al. (2020) [39]: *Scyliorhinus duhamelii* and *Squatina aculeata*.

In 2022, Soldo and Lipej presented the most recent checklist of Adriatic chondrichthyans [38]. It includes the following species that we have not been able to confirm for Croatian waters: *Raja undulata*, *Rhinoptera marginata*, *Rhinobatos rhinobatos*, *Somniosus rostratus* (Risso, 1827), *Rhizoprionodon acutus*, *Galeus atlanticus* (Vaillant, 1888), and *Hexanchus nakamurai* [38]. The potential presence of these species in Croatian waters remains to be verified in the future.

We have been able to confirm the presence of two entirely new species for this area that were neither included in the Croatian checklist from 2009 [17] nor in the checklists for the Adriatic Sea from 2020 [39] and 2022 [38]: *Scyliorhinus duhamelii* and *Squatina aculeata* (see discussion below). The verified record of the latter species in our study is the first documented one for Croatian waters as well as the Adriatic Sea. Furthermore, our record of *Alopias superciliosus* is the first of this species from Croatian waters.

### 4.2. Chondrichthyan Distribution Patterns in Croatia

A heatmap of all records of chondrichthyan species in Croatia collected during this study is provided (Figure 3), which displays the combined records for all species that were identified in Croatian waters. Hotspots are recognizable in the main harbors of Croatia, namely Rijeka, Split and Dubrovnik, but also in the Kvarner Bay and the Brač channel. Note that these areas are not covered by Marine Protected Areas (Figure 4). However, these hotspots may represent a bias in data collection since these are also the main landing areas of Croatia and may thus have contributed to a disproportionally high number of records. Recent citizen science records are mostly from the Kvarner Bay in the north of Croatia due to the frequency of tourism in this area. The lack of more recent records from regions in the south does not necessarily mean that there are more individuals in the north, but rather that tourism is not as prevalent in the south as it is in the north of Croatia. Future studies should therefore focus on southern Croatian waters.

### 4.3. Extended Discussion on Chondrichthyan Species in Croatia

#### 4.3.1. Chimaeriformes

*Chimaera monstrosa* Linneaus, 1758 is the only holocephalan species in our dataset and is the only known chimaeriform known from the Adriatic Sea [4]. The IUCN status of *C. monstrosa* is near threatened in the Mediterranean [12]. *Chimaera monstrosa* was documented by two historical records from 1962 and 1996 only, which were from the deeper waters of south Croatia, caught at a depth of approximately 500 m.

#### 4.3.2. Rajiformes

Skates are a highly diverse group with nearly 300 species worldwide and are quite abundant in the Mediterranean Sea [4]. Four genera with eighteen species occur in this region, of which two species are assumed to be endemic [4]. We can confirm ten species occurring in Croatian waters.

*Dipturus* cf. *batis* (Linnaeus, 1758) is considered critically endangered globally by the IUCN Red List but has not been evaluated for the Mediterranean Sea so far [12]. According to recent studies, the common skate *D. batis* is a species complex that tends to be separated into two nominal species: The blue skate (formerly called *Dipturus* cf. *flossada* (Risso, 1826; now called *D. batis*)) and the flapper skate (*D. intermedius* (Parnell, 1837)). They are primarily distinguishable by the coloration of their irises, and have different clasper lengths at maturation, which is why it is assumed that no hybridization between these two species is possible [81]. Although we have photos of one specimen from Rijeka, we have not been able to clearly identify it as either one of the two species. Identification literature is mostly based on adult male specimens according to the presence or absence of malar thorns and on coloration [81], but this is a female specimen that is conserved in alcohol and has thus lost its coloration. Therefore, we cannot be sure if this and the other four records are *D. batis* or *D. intermedius*, so we state them here as *Dipturus* cf. *batis*. This is a good example of the need for a thorough taxonomic revision of historic as well as recent specimens, especially considering their respective geographic ranges in the Mediterranean Sea and their IUCN status [12,82]. Future studies should investigate whether the species is currently present in the area and should also verify the documentation of specimens recorded in the past [82]. This species is rarely reported and is included in the Croatian list of strictly protected species (Regulation on strictly protected species, Annex 1, 2013) [83].

*Dipturus nidarosiensis* (Storm, 1881) is listed as near threatened globally by the IUCN and has not been evaluated for the Mediterranean Sea [12]. Two individuals were caught in deeper waters in the very south of the Adriatic Sea, at the southern margin of Croatia’s Exclusive Economic Zone, during the FAO AdriaMed Deep Sea Expeditions in 2008 and 2010 [32].

*Dipturus oxyrinchus* (Linnaeus, 1758) is considered near threatened in the Mediterranean Sea by the IUCN [12]. We have collected numerous records of this species over the past hundred years, all of which are from the far south, near Dubrovnik. It occurs in the southern and central Adriatic Sea [34,38,84] and is included in the Croatian list of strictly protected species (Regulation on strictly protected species, Annex 1, 2013) [83].

*Leucoraja circularis* (Couch, 1838) is critically endangered in the Mediterranean Sea [12]. It only occurs in the southern Adriatic Sea at the continental shelf-slope boundary [24,85]. This species has rarely been reported up until now.

*Leucoraja fullonica* (Linnaeus, 1758) is critically endangered in the Mediterranean Sea [12]. Croatian records are based on one juvenile specimen caught near Split [34]. It is also reported from the southern Adriatic Sea [35].

*Raja asterias* Delaroche, 1809 is considered near threatened in the Mediterranean Sea by the IUCN [12]. It is considered common in the Adriatic, but it is more abundant in the west than in the east, where the Croatian coast lies [38].

*Raja clavata* Linnaeus, 1758 is considered near threatened in the Mediterranean Sea by the IUCN [12]. This species is very common in Croatian waters and is regularly sold on fish markets [86]. The most recent records from 2019 to 2021 are contributions from citizen science and are from Krk and Rovinj in the far north, as well as Molunat and Lastovo in the far south, which also corresponds with its general distribution along the entire Croatian coastline.

*Raja miraletus* Linnaeus, 1758 is considered to be of least concern in the Mediterranean Sea by the IUCN [12]. It is widespread throughout Croatian waters and commonly found at fish markets [86]. It is abundantly found in bottom trawl catches [87].

*Raja polystigma* Regan, 1923 is considered to be of least concern in the Mediterranean Sea by the IUCN [12]. Recently (19 July 2022), one specimen was caught near the Island of Ist (Figure 1D) and was identified by the diffuse yellow circle around a dark-centered ocellus on each pectoral fin [4]. The species *R. montagui* and *R. polystigma* look very similar, which leads to problems in their correct identification [82]. Recent genetic studies support a separation of the two species and their distribution in the Mediterranean Sea: *Raja montagui* is limited to the North African coasts of Algeria and Tunisia, while *R. polystigma* is endemic to the Mediterranean Sea and is distributed along its entire coast [88]. Although we had access to alleged specimens of both species at the Natural History Museum Vienna, we were not able to re-identify them since preservation in alcohol had bleached their coloration, obscuring any useful morphological discriminatory traits.

*Raja montagui* Fowler, 1910 is considered to be of least concern in the Mediterranean Sea, according to the IUCN [12]. This species is regularly reported in Croatian waters. However, as described above, *R. montagui* is difficult to distinguish from the closely related *R. polystigma*. Therefore, further research is needed in order to confirm the records of *R. montagui* in Croatia and to better differentiate the two species in the future.

*Raja radula* Delaroche, 1809 is endangered in the Mediterranean Sea, according to the IUCN [12]. It was reported from trawling fisheries along the entire Croatian coastline [29,38]. Some specimens were also found near Rijeka in the northern Adriatic [29].

*Raja undulata* Lacepède, 1802 is considered near threatened in the Mediterranean Sea, according to the IUCN [12]. This species was listed in the last checklist from 2009 [17], but recently its distribution in the Adriatic has been questioned. The only specimen reported by Ninni (1912) from the Museo Civico di Storia Naturale di Venezia has doubtful coloration and no other visible evidence to confirm the identification [39]. Therefore, more evidence is needed to assess the historical and current presence of this species in Croatia.

*Rostroraja alba* (Lacepède, 1803) is listed as endangered by the IUCN Red List in the Mediterranean Sea [12]. It is considered very rare in the Adriatic Sea and is reported only sporadically from the central and southern Adriatic areas [34]. However, only recently were two specimens found at the fish market in Split, one of which was caught near Šćedro Island.

#### 4.3.3. Myliobatiformes

Stingrays and their relatives (Myliobatiformes) are morphologically the most diverse group of any ray order [4]. In Croatian waters, they are represented by only seven species.

*Bathytoshia lata* (Garman, 1880) is listed as vulnerable globally by the IUCN, but its status in the Mediterranean Sea has not been assessed yet [4,12]. This species was originally described as *Trygon lata* Garman, 1880 and is distributed from Europe to Australia [4,89]. For some time, there was confusion as it was previously reported as *B. centroura* (Mitchill, 1815) as well as its synonym *Dasyatis centroura* (Mitchill, 1815), but this species is now considered a separate one that only occurs in North and South America [4,89]. *Bathytoshia lata* can be found in the southern and central Adriatic areas, mostly as bycatch from trawling [38]. It is considered very rare in the Mediterranean Sea [82]. In 2002, the largest specimen of this species ever documented was caught near Koločep Island [40].

*Dasyatis pastinaca* (Linnaeus, 1758) is listed as vulnerable by the IUCN in the Mediterranean Sea [12]. There is continuous historical evidence for *D. pastinaca* in Croatian waters, with many recent citizen science reports. It is a common species found in all Croatian waters, especially in the channel areas [34,38]. It is included in the Croatian list of strictly protected species (Regulation on strictly protected species, Annex 1, 2013) [83].

*Pteroplatytrygon violacea* (Bonaparte, 1832) is listed as of least concern in the Mediterranean Sea by the IUCN [12]. All records are relatively recent; ten of them are from citizen science, mostly collected by recreational fishermen or seen by beachgoers. This species has only been found in northern Croatian waters along the Istrian coastline and in the Kvarner Gulf. This distribution corresponds to previous studies from the northern Adriatic Sea [90]. One specimen was reported to have swum up the Lim channel, indicating a temporary tolerance for brackish waters (45.130085° N, 13.736721° E).

*Gymnura altavela* (Linnaeus, 1758) is considered critically endangered by the IUCN in the Mediterranean Sea [12]. It is a very rare species in Croatian waters, but it has been reported occasionally [40]. Two historical records are from 1927. In 2022, a specimen was accidentally caught off the Duilovo coast near Split using a longline at 15 m depth, indicating the continuous presence of this species in Croatian waters. This specimen weighed approximately 10 kg and had a disc width of 150 cm. It is included in the Croatian list of strictly protected species (Regulation on strictly protected species, Annex 1, 2013) [83].

*Aetomylaeus bovinus* (Geoffroy Saint-Hilaire, 1817) is critically endangered both globally and in the Mediterranean Sea, according to the IUCN [12]. It was previously recorded throughout southern Croatian waters and recently in the northern Adriatic Sea in the Gulf of Trieste [91]. The new citizen science records included here show a similar distribution.

*Myliobatis aquila* (Linnaeus, 1758) is listed as vulnerable in the Mediterranean Sea by the IUCN [12]. This species is very common in Croatian waters, with many historical as well as recent occurrences.

*Mobula mobular* (Bonnaterre, 1788) is endangered in the Mediterranean Sea, according to the IUCN Red List [12], and is strictly protected in Croatian waters by several international conventions [4]. The work of Fortuna and colleagues [92] on its occurrence in the central and southern Adriatic suggests that this species is probably much more common than previously thought. Observations by citizen scientists support that hypothesis. Four recent (2019–2021) citizen science records from Žirje were collected in August and September, indicating a potential seasonality in their occurrences. It is included in the Croatian list of strictly protected species (Regulation on strictly protected species, Annex 1, 2013) [83].

#### 4.3.4. Torpediniformes

Four species of torpedo rays occur in the Mediterranean Sea, three of which we identified in Croatian waters.

*Tetronarce nobiliana* (Bonaparte, 1835) is listed as of least concern globally by the IUCN, but its status in the Mediterranean Sea has not been assessed yet [12]. It is rarely reported from the central and southern Adriatic Seas, and so far only juvenile specimens have been reported [38,39]. The single specimen we identified was caught on 17 April 2018, south of Slano by bottom trawling at 250 m depth and weighed 17.5 kg (Figure 1B).

*Torpedo marmorata* Risso, 1810 is considered to be of least concern in the Mediterranean Sea by the IUCN [12]. We have found numerous records from the last two hundred years that show that this species is widely distributed across Croatia’s Exclusive Economic Zone and coast. Scuba divers often encounter the marbled electric ray in shallow water.

*Torpedo torpedo* (Linnaeus, 1758) is considered to be of least concern in the Mediterranean Sea by the IUCN [12]. We were able to verify one record from Split housed in the Natural History Museum of Vienna (Figure 1B), and photos of this specimen were taken during this study. The other record is from the MEDITS survey [33]. The rare occurrences of this species in the Adriatic Sea may be explained by its preference for warmer, tropical waters [4]. Recently, more specimens were found in Albania by Andej Gajić and in the waters off Montenegro by Ilija Četković (pers. comm.), suggesting that the species is more common in the southern Adriatic Sea.

#### 4.3.5. Rhinopristiformes

Shovelnose rays and their closest allies are a diverse and characteristic group that has experienced significant declines in their populations [4].

*Pristis pectinata* Latham, 1794 is considered critically endangered in the Mediterranean Sea by the IUCN [12] and is even assumed to be potentially extinct in this area [42]. The youngest occurrence record we can confirm comes from 1929 and is housed in the Prirodoslovni Muzej Split. All records found in this study are from southern Croatia. The population of the smalltooth sawfish was never very large in the Mediterranean Sea, and it was further depleted by extensive fishing [42]. The last two checklists for the Adriatic Sea consider this species to be regionally extinct [38,39]. It is included in the Croatian list of strictly protected species (Regulation on strictly protected species, Annex 1, 2013) [83].

*Rhinobatos rhinobatos* (Linnaeus, 1758) is listed in the Croatian checklist from 2009 but was excluded from the Adriatic Sea checklist by Kovačić and colleagues [39]. We have not been able to find any past or present evidence for this species. Thus, its presence in Croatian waters has to be considered ambiguous at best. It is included in the Croatian list of strictly protected species (Regulation on strictly protected species, Annex 1, 2013) [83].

#### 4.3.6. Hexanchiformes

The cow and frill sharks mostly occur in water depths between 100 and 2500 m [4].

*Heptranchias perlo* (Bonnaterre, 1788) is listed as data deficient by the IUCN in the Mediterranean [12]. Until recently, this species was only known in the Adriatic Sea from a few anecdotal reports [38,52]. In 1948, it was caught in the Adriatic Sea during the Hvar trawl survey, but no exact location was provided [87]. However, more recently, this species has been caught in Albanian waters [93] and off Slovenia [38]. Also, Dragičević and Isajlović reported two recently caught individuals (2020 and 2022) in Croatian waters [43]. It is included in the Croatian list of strictly protected species (Regulation on strictly protected species, Annex 1, 2013) [83].

*Hexanchus griseus* (Bonnaterre, 1788) is considered to be of least concern in the Mediterranean Sea by the IUCN [12]. The records for the bluntnose six-gill shark show a distribution along the entire Croatian coastline. The latest records are from the Zadar region in the central part of Croatia’s sea, whereas the historical records are from the far north and the far south. This species is regularly reported from the Adriatic Sea [94]. It is included in the Croatian list of strictly protected species (Regulation on strictly protected species, Annex 1, 2013) [83].

#### 4.3.7. Echinorhiniformes

*Echinorhinus brucus* (Bonnaterre, 1788) is endangered globally and in the Mediterranean Sea, according to the IUCN [12]. The only record from Croatia is from the Kvarner Gulf, which was reported on 5 May 1877, and this taxidermized specimen is housed in the Venice Museum of Natural History, Fontego dei Turchi, under inventory number 7781 (Figure 1E). This species is mostly found at water depths between 200 and 900 m, which may be the reason why very little is known about its historic and current populations in the Mediterranean Sea [4].

#### 4.3.8. Squaliformes

This group is the second-most speciose shark order in the world, and nine species are known to occur in the Mediterranean Sea [4], with five species being present in Croatian waters. Although dogfish sharks inhabit a wide range of habitats from shallow to deep waters, they are known to dominate benthic shark communities at depths of over 3000 m [4]. Since deep sea explorations are rarely performed and the maximum depth of the Adriatic Sea is only 1233 m, not much is known about this order in this region.

*Squalus acanthias* Linnaeus, 1758 is considered endangered in the Mediterranean Sea by the IUCN [12]. We have found continuous records over the last 200 years. It occurs along the entire Croatian coast but is more common in the northern part and in the channels between the islands [34].

*Squalus blainville* (Risso, 1827) is considered data deficient both globally and in the Mediterranean Sea by the IUCN [12]. The longnose spurdog has been found in central and southern Croatia [34]. Eighteen individuals were caught during the MEDITS survey alone [33].

*Centrophorus* cf. *uyato* (Rafinesque, 1810), is endangered globally and critically endangered in the Mediterranean Sea, according to the IUCN [12]. In Croatian waters, it has only been recorded in the deeper waters of the south Adriatic pit [24,29]. It was previously referred to as *C. granulosus* (Bloch and Schneider, 1801), but recent studies have shown that Mediterranean individuals of this genus are molecularly and morphologically distinct from individuals from the Atlantic and Indian Oceans [82,95]. Therefore, the Mediterranean resident species should now be considered *C.* cf. *uyato* [82,95].

*Etmopterus spinax* (Linnaeus, 1758) is listed as least concern by the IUCN [12]. Most records we identified for this species are from Kirincić and Lepetić [24] and are all coming from the same locality near Dubrovnik. The other historical records correspond with this southern distribution [29]. This species predominantly occurs in deeper waters, which may explain the lack of recent citizen science records as well as the southern distribution near the south Adriatic pit.

*Oxynotus centrina* (Linnaeus, 1758) is critically endangered in the Mediterranean Sea, according to the IUCN [12]. Records show a distribution along the entire Croatian coast, with a particular aggregation near the central Adriatic Sea, where many new individuals have been found in recent years [48]. Almost all records are from the last twenty years when the sharks were caught by trawling in water depths between 40 and 125 m. It is included in the Croatian list of strictly protected species (Regulation on strictly protected species, Annex 1, 2013) [83].

*Dalatias licha* (Bonnaterre, 1788) is listed as vulnerable in the Mediterranean by the IUCN Red List [12]. One record includes a specimen reported in Croatia’s Exclusive Economic Zone [29], but no new records have been reported from Croatia since then.

#### 4.3.9. Squatiniformes

Three species of Squatinidae have been recorded from the Mediterranean Sea, all of which are considered critically endangered by the IUCN globally as well as in the Mediterranean Sea [12]. Of these, *Squatina squatina* (Linnaeus, 1758) is the most common species. The other two species have been very difficult to confirm in Croatian waters due to scarce records until now.

*Squatina aculeata* Cuvier, 1829 is critically endangered globally as well as in the Mediterranean Sea, according to the IUCN [12]. An alleged specimen of the sawback angelshark, which was caught in April 1939 in the Split area, was reported by Holcer and Lazar [49]. It is preserved as an 80-centimetre-long dermoplastic and is listed as *Squatina fimbriata* Müller & Henle, 1839 in the collection catalogue of the Croatian Natural History Museum in Zagreb (inventory number 3348) [49]. Unfortunately, this specimen could not be found in the collection, and thus its correct identification cannot be verified by us.

However, we found one specimen of *S. aculeata* in the Natural History Museum Vienna (inventory number NHMW—88925). It is represented by a 26-centimetre-long juvenile that was collected by the notable Croatian ichthyologist Dr. Juraj Kolombatović (1843–1908); according to the label, the location where this specimen was collected is Split. Characteristic for this species is the position of the first dorsal fin, whose origin lies anteriorly to the posterior tips of the pelvic fins (Figure 5C). The specimen is scattered with small dark spots, and no large ocelli are visible (Figure 5A). The width of the pectoral fins of this specimen is about half their length and not two-thirds as wide as would be the case in *S. squatina* (Figure 5B). However, this ratio might have been altered during preservation. The nasal barbels and anterior nasal flaps are heavily fringed, with no straight or spatulate tip present (Figure 5D,E). Only weak lobes are present at the lateral head folds. A weakly concave interorbital space is visible. In addition, large, prominent thorns are present along its midback line, which is the main characteristic of *S. aculeata*. However, small thorns along the midback can be present in juveniles of *S. squatina* as well [4,96]. No prominent thorns are present on the snout and between the eyes, although all *Squatina* species have them when they are adults, so this may be because this specimen is a juvenile. Since very little is known about the ontogeny of angelsharks, we do not know how traits are expressed at different life stages, especially in juveniles. Nonetheless, the traits described above match those described for *S. aculeata* [4]. Therefore, this specimen confirms the historical presence of this species in the Adriatic Sea. Distinguishing between *S. squatina* and *S. aculeata* based on morphology alone has always been difficult, especially in juveniles, but genetic analyses may be a promising tool to complement morphological observations in the future [97].

Currently, the known range of this species does not include the Adriatic Sea but rather the Aegean Sea, as well as Sicily, Algeria and Tunisia [96]. Historically, however, this species was widely distributed across the entire Mediterranean Sea, although it has only been sparsely reported up to now [96]. Since this specimen and the lost specimen from Zagreb are quite old, a historical distribution of this species in the Adriatic Sea can be confirmed. Nowadays, however, the presence of this species in the Adriatic Sea is uncertain due to decades of overfishing [98]. It is highly likely that it is now regionally extinct.

*Squatina oculata* Bonaparte, 1840 is critically endangered globally as well as in the Mediterranean Sea, according to the IUCN [12]. Holcer and Lazar [49] found a supposed specimen of *Squatina oculata* in the collection catalogue of the Croatian Natural History Museum Zagreb (inventory number 2539). This female specimen from Bakarac was caught on 21 July 1893. However, the identity of *S. oculata* on the label could not be verified, according to the authors [49]. This specimen could not be found in the collection, and therefore its identification remains unverified.

We found two embryos of *S. oculata* in the Natural History Museum of Vienna (inventory number NHMW—22429). These are labelled with the date of 6 January 1888, and were sent by Dr. Kolombatović from Split, where they had presumably been retrieved from their gravid mother. The specimens were reported as *Rhina fimbriata* and are still attached to their yolk sacs. The position of the first dorsal fin is striking since it is well posterior to the posterior tips of the pelvic fin, which is the main diagnostic character of this species [4] (Figure 6E). In addition, the width of the pectoral fins is about half their length, and large symmetrical dark blotches are visible on the pectoral fins as well as on the tail (Figure 6A). The body is dotted with small dark spots and symmetrical larger white spots (Figure 6). No thorns are present along their midback lines or on the snout and between the eyes, which may develop later in adulthood (Figure 6B,C). A concave interorbital space is present (Figure 6D). The lateral head folds do not have lobes (Figure 6D). Whether or not the nasal barbels and anterior nasal flaps are weakly fringed cannot be assessed due to their poor preservation. Although identification of embryos is difficult because the identification literature we used is based on adults [4] and traits are likely to change over the lifetime of these animals, we are confident that these two embryos are indeed *S. oculata*. The position of the first dorsal fin, which is well posterior to the pelvic fin, is unique to this species. Therefore, these two embryos confirm the presence of *S. oculata* in the Adriatic Sea, at least historically.

Currently, the known range of *S. oculata* includes the Aegean Sea, Turkey, Tunisia, Sicily and the southern Ionian Sea, but not the Adriatic Sea [96]. This species was listed in the last checklist for Croatia published in 2009, based on its occurrence in the Adriatic Sea mentioned by Brusina in 1888 [17,57]. Severe overfishing has afflicted this species too, which may be why it has not been seen in Croatian waters since the 19th century [98], indicating its local extinction. It is included in the Croatian list of strictly protected species (Regulation on strictly protected species, Annex 1, 2013) [83].

*Squatina squatina* (Linnaeus, 1758) is critically endangered globally as well as in the Mediterranean Sea, according to the IUCN [12]. It used to be very common in Croatian waters and has been frequently recorded over the last 100 years [99]. This species was widespread throughout the entire Croatian Sea, but recent data suggests that it is now present exclusively in a small area near the Molat archipelago. It is included in the Croatian list of strictly protected species (Regulation on strictly protected species, Annex 1, 2013) [83].

#### 4.3.10. Lamniformes

Almost all species of mackerel sharks that occur in the Mediterranean Sea have also been found to occur in Croatian waters. Only *Isurus paucus* Guitart, 1966 has not been reported yet. The presence of the sandtiger shark, *Carcharias taurus*, however, remains ambiguous (see discussion below). The two species *C. taurus* and *Odontaspis ferox* (Risso, 1810) are very hard to differentiate without examining the tooth morphologies, although they belong to two entirely separate families (namely Carchariidae and Odontaspididae), based on both molecular and morphological evidence [100].

*Alopias superciliosus* Lowe, 1841 is endangered in the Mediterranean Sea, according to the IUCN [12]. One individual was caught with a longline near Dubrovnik in 2017 (Figure 1C), representing the first record from Croatian waters.

*Alopias vulpinus* (Bonnaterre, 1788) is endangered in the Mediterranean, according to the IUCN [12]. We have found one historical record from 1890 and three citizen science records from 2020 and 2021. The historical record is from Dubrovnik; the new ones are from further north along the coast. This species is common in Croatian waters, especially in the northern and central Adriatic [34]. Recently, fishermen noticed a decline in catches [38]. It is included in the Croatian list of strictly protected species (Regulation on strictly protected species, Annex 1, 2013) [83].

*Carcharias taurus* Rafinesque, 1810 is considered critically endangered both globally and in the Mediterranean Sea, according to the IUCN [12]. It is included in the Croatian list of strictly protected species (Regulation on strictly protected species, Annex 1, 2013) [83]. There are a few unconfirmed records from the Adriatic Sea that are likely the result of misidentifications. In Croatian waters, we were able to find two historical records of *C. taurus* from the 19th century [53,57,58], but without any photographic evidence available. According to Brusina (1888), one specimen, which was caught in 1881, is stored in the Museo di Storia Naturale in Trieste [57]. We were, however, not able to obtain access to this specimen and are therefore not able to reliably verify the presence of this species in Croatian waters. One additional record from post-1970 is based on personal communication alone and can thus not be verified either [55]. Allegedly, the head of a sand tiger shark was found at a fish market in Split in 2002 and then stored at the Institute of Oceanography and Fisheries in Split [52]. However, there are no documented records of this head at this institute, and we were not able to locate the whereabouts of the head.

Only one record of *C. taurus* from Croatian waters was published with photographic evidence [51,52]. This specimen was caught off the island of Molat in September 1999. However, a closer examination of the photograph and an additional photo of the same specimen (Figure 7) raise doubts about its correct identification. The specimen in the photo has a uniform coloration, whereas *Carcharias taurus* usually has brown spots scattered across its body [4]. Also, the position of the first dorsal fin originates over the rear tips of the pectoral fin, while in *C. taurus,* the first dorsal fin originates closer to the pelvic fin than to the pectoral fin [4]. Furthermore, the snout is conical and relatively long, whereas it is short in *C. taurus*, and the eyes are large but small in *C. taurus*. All of the characters stated above are diagnostic for the smalltooth sandtiger shark *Odontaspis ferox* and question previous identifications of the sandtiger shark *Carcharias taurus* in the Adriatic Sea.

*Carcharodon carcharias* (Linnaeus, 1758) is critically endangered in the Mediterranean Sea, according to the IUCN [12]. The white shark has a wide distribution across Croatia’s sea and has had continuous records over the last 100 years, including numerous attacks on humans [62,66,67,68]. The last record was also an attack that happened in 2008 [67,68]. Although white sharks have been regularly encountered over the last decades, our dataset suggests that the number of reports has decreased since the latter half of the 19th century (Appendix A). The white shark is included in the Croatian list of strictly protected species (Regulation on strictly protected species, Annex 1, 2013) [83].

*Cetorhinus maximus* (Gunnerus, 1765) is listed as endangered in the Mediterranean Sea by the IUCN [12]. The basking shark is the only plankton-feeding shark species occurring in Croatian waters. It has a continuous record over the last 200 years, and is distributed across the entire Croatian Sea. It is included in the Croatian list of strictly protected species (Regulation on strictly protected species, Annex 1, 2013) [83].

*Isurus oxyrinchus* Rafinesque, 1810 is considered critically endangered in the Mediterranean Sea by the IUCN [12]. This species is widely spread across the entire Croatian shoreline. Most records for this species are from the late 19th century [57]. We have found only scarce records from the 20th century, culminating in the three newest records from 2015 to 2019 [70]. Newer records have been presented in social networks from citizen science but still have to be verified by scientists. It is included in the Croatian list of strictly protected species (Regulation on strictly protected species, Annex 1, 2013) [83].

*Lamna nasus* (Bonnaterre, 1788) is also critically endangered in the Mediterranean Sea, according to the IUCN [12]. There are many historical records of this species [38]. We found only one definite but very recent record from 2018 near the Channel of Split [72]. Additionally, there are unpublished records [38] and information on social media networks that need to be verified in the future. It is included in the Croatian list of strictly protected species (Regulation on strictly protected species, Annex 1, 2013) [83].

*Odontaspis ferox* (Risso, 1810) is considered vulnerable globally and critically endangered in the Mediterranean Sea, according to the IUCN [12]. It is included in the Croatian list of strictly protected species (Regulation on strictly protected species, Annex 1, 2013) [83]. For the possible confusion of the specimen from the island of Molat from September 1999 with *C. taurus,* please see the discussion above. Although few historical records date back prior to 1960 [101], we have found one more record of *Odontaspis ferox* from Stari Grad on the island of Hvar from 1911 (Figure 8) [50]. Here, the position of the first dorsal fin is clearly closer to the pectoral fin than the pelvic fin. Also, the first dorsal fin is much larger than the second dorsal and anal fins.

Only one other record of *O. ferox* has been documented, which is from 1954 [30]. This specimen is mentioned in the collection catalogue of the Dubrovnik Natural History Museum from 1989 but has been lost since, probably during the Croatian War of Independence. This specimen was from the island of Lokrum, just off Dubrovnik. Still, with the records from Molat and Hvar, we can confirm the presence of the smalltooth sandtiger shark in Croatia. *Carcharias taurus*, however, cannot be confirmed as of now and remains doubtful.

#### 4.3.11. Carcharhiniformes

Six families, eleven genera and thirty-two species of ground sharks occur in the Northeast Atlantic and the Mediterranean Sea [4]. In Croatian waters, five families and eleven species represent this order.

*Galeus melastomus* Rafinesque, 1810 is listed as of least concern in the Mediterranean Sea by the IUCN [12]. It is a very common deep-water species that is mainly distributed in the south Adriatic pit [24]. This deep-water lifestyle explains why this species is unlikely to be observed by divers, snorkelers and recreational fisherpersons.

*Scyliorhinus canicula* (Linnaeus, 1758) is considered of least concern by the IUCN [12]. It is a very common species that is distributed along the entire Croatian coastline. Forty-seven records are from Kirincić and Lepetić alone [24], and one record from the MEDITS bottom trawl survey found two hundred and eighteen individuals alone [33]. All recent citizen science records are from Istria and were collected by scuba divers in shallow waters. It may be possible that some historical records of this species are in fact *S. duhamelii*, but more research is needed to clarify this issue (see the discussion below).

*Scyliorhinus duhamelii* (Garman, 1913) is not listed in the IUCN Red List, probably due to the taxonomic controversy surrounding this species [12]. The validity of this species was under discussion [82], as it is morphologically very similar to *S. canicula* and co-occurs in the same area [102,103]. The species is now considered valid by WoRMS and Fishbase [104,105]. Over the last decades, these two species were regarded as synonyms but are now considered sister taxa [102]. The lectotype of *S. duhamelii* is from the Adriatic Sea, and all other specimens accordingly described are from the Mediterranean Sea, namely Croatia, Morocco and Algeria (see Soares and Carvalho 2019, Figure 45 [103]). *Scyliorhinus duhamelii* is differentiated from *S. canicula* based on: (1) the pattern of its scattered dark spots in varied sizes, which form aggregations (see Figure 1A); (2) the lateral positions of the shallow nasoral grooves and posterior nasal flaps; (3) the size of the lower labial furrow [102,103]; and (4) the dorsal fins are situated more posteriorly than in the other two catshark species [102,103]. Further morphological differences can be found in the intestine valve and the claspers [102,103]. Differences in sexual organs are informative for phylogenetic relationships in the genus *Scyliorhinus* and may be an indicator that a speciation event occurred [102]. In our study, all records of *S. duhamelii* were recently collected by citizen scientists in northern Croatia, in Molat and Rab. The lack of historical data can be explained by the synonymization of *S. duhamelii* and *S. canicula* for so many years rather than their previous absence. Still, it might be possible that these two species are two morphotypes of the same species: *S. duhamelii* with white spots and *S. canicula* without white spots, as can be seen on Plate 1 in Appendix II of Schipany and Kriwet 2018 [106]. There is also the possibility that *S. duhamelii* hybridizes with *S. canicula*. It was also suggested that *S. duhamelii* may be endemic to the Mediterranean Sea [4], but further, especially molecular, studies [82] are necessary to assess whether this species is taxonomically valid in this area and not just a colour morph of *S. canicula*.

*Scyliorhinus stellaris* (Linnaeus, 1758) is listed as near threatened in the Mediterranean Sea by the IUCN [12]. It has a similar distribution as *S. canicula*. According to Soldo and Lipej’s checklist, it occurs in large numbers in the entire Adriatic Sea except for the Gulf of Trieste, where it is considered rare [38]. Dulčić and Kovačić, however, state that nowadays it is only common around the islands near Zadar [34]. Our recent citizen science records were collected mostly by scuba divers in the Kvarner Bay, Velebit Channel and western Istria near Rovinj, demonstrating a more northern distribution of this species.

*Galeorhinus galeus* (Linnaeus, 1758) is critically endangered globally and vulnerable in the Mediterranean [12]. It is occasionally recorded throughout the Adriatic Sea. Most records are from the far south of Croatia, near Dubrovnik. The youngest record from 1992, however, is from far up north in Urinj. It is included in the Croatian list of strictly protected species (Regulation on strictly protected species, Annex 1, 2013) [83].

*Mustelus asterias* Cloquet, 1819 is vulnerable in the Mediterranean Sea, according to the IUCN [12]. It occurs from the middle of Croatia, near Zadar, to the south of Croatia. Nearly all catches of this species were performed during trawl surveys [33,73,74,75,101]. We have found only seven records in the last seventy years. It used to be found throughout the entire Adriatic Sea, but now it is mainly distributed in the southern and central Adriatic Seas and is considered rare in the northern part [34].

*Mustelus mustelus* (Linnaeus, 1758) is considered vulnerable in the Mediterranean Sea by the IUCN [12]. It is common in Croatian waters but more numerous in the northern Adriatic Sea and channel areas [34]. The records that we found show that it is scattered around most of Croatia, with a denser distribution in the south but no occurrences in the Kvarner Bay. It has a continuous record from the late 19th century until now (Table 1 and Appendix A). This species is targeted by commercial and recreational fisheries and can thus be regularly seen in fish markets.

*Mustelus punctulatus* Risso, 1827 is considered vulnerable in the Mediterranean by the IUCN [12]. In spite of that, this species is common in Croatian waters but more numerous in the channel areas [34]. It was mostly caught by longlines or during trawl surveys and has a similar distribution as the common smooth-hound shark. We have found only a few, but continuous, records over the last ninety years. This species is also targeted by commercial and recreational fishermen and can regularly be found in fish markets.

*Carcharhinus plumbeus* (Nardo, 1827) is listed as endangered in the Mediterranean by the IUCN [12]. Evidence for the presence of the sandbar shark in the Adriatic Sea is mainly based on records of newborn and juvenile specimens, especially in the northern Adriatic Sea, which has led some authors to conclude that the Adriatic Sea may serve as a nursery area for this shark [38]. The sandbar shark has a southern distribution from Zadar to Dubrovnik, which is confirmed by historical literature as well as citizen science data [107]. It is included in the Croatian list of strictly protected species (Regulation on strictly protected species, Annex 1, 2013) [83].

*Prionace glauca* (Linnaeus, 1758) is critically endangered in the Mediterranean Sea, according to the IUCN [12]. It can be found throughout the entire Croatian Exclusive Economic Zone, from the coastal areas to the open waters. We have collected eleven recent citizen science records from 2020 and 2021 for this species. It is included in the Croatian list of strictly protected species (Regulation on strictly protected species, Annex 1, 2013) [83].

*Sphyrna zygaena* (Linnaeus, 1758) is listed as critically endangered in the Mediterranean Sea by the IUCN [12]. All records of the only hammerhead species in Croatian waters are from historical data, with the last record reported in 1959. As this species has always been very rare in the Adriatic Sea, it can be assumed that it is a vagrant species. It is included in the Croatian list of strictly protected species (Regulation on strictly protected species, Annex 1, 2013) [83].

*Sphyrna tudes* (Valenciennes, 1822) is only mentioned in reports from the 19th century, with records that cannot be unambiguously confirmed and were subsequently doubted [39]. Therefore, this species is excluded from the list presented here.

## 5. Conclusions

Our examination of numerous data sources revealed the unambiguous presence of 54 chondrichthyan species in Croatian waters. These include 30 shark species, 23 ray and skate species, and one chimaeroid species. Five of these species are reported in this area for the first time: *Dipturus nidarosiensis*, *Raja montagui*, *Squatina aculeata*, *Alopias superciliosus*, and *Scyliorhinus duhamelii*.

The methods of data collection used in this study have both advantages and disadvantages. Historical data, for example, poses significant problems in identification due to poor preservation, such as fading colours and changes in the shape of the body, e.g., skin wrinkles. However, museum specimens provide some of the oldest non-fossil records of sharks, rays, skates, and chimaeras in this area.

Data from the literature may be cited several times in different publications, so errors in the transcription of the data may have occurred. One example is the size of the *Echinorhinus brucus* specimen from the Kvarner Gulf (Figure 1E), which is reported to be 162 cm in Trois (1876), but 145 cm in Mizzan (1994) [45,46].

Citizen science data from the MECO Project and WWF Adria have the advantage that they provide accurate and recent accounts of sighted species due to the fact that only records with photographic evidence were collected and then identified by scientists. These animals were either photographed alive or freshly killed, so morphological traits were well visible. One disadvantage of citizen science, however, is the geographical concentration of data collected in touristic regions, and consequently, regions with a lack of tourism are undersampled. These include regions that are difficult to reach by the general public due to environmental factors such as shores with steep cliffs, areas with turbulent water, or the deep sea. These sampling biases must be carefully taken into account when analyzing citizen science data.

Our analyses indicate that threatened species are more likely to be encountered by fishermen than by recreational divers or snorkelers (Figure 2C,D). Overall, only 16 out of 54 species (30%) were found by scuba divers and snorkelers. The lack of sightings by snorkelers indicates that these species are very rare nowadays, particularly in shallow waters and touristic areas. Nonetheless, in combination with fishing, citizen science provided evidence for 24 out of all 54 species (44%) in Croatian waters, representing almost half of all occurring species. In this study, citizen science records significantly support the evidence collected from museums and published records. Many species on this checklist could only be accurately verified because citizen scientists provided photographs. Four species even have records exclusively provided by citizen science, with no historical evidence found so far. These species are *Alopias superciliosus*, *Mobula mobular*, *Scyliorhinus duhamelii* and *Tetronarce nobiliana.* This shows the potential of citizen science to detect rare species, add to our knowledge of occurrences, and establish local and regional diversity patterns. We conclude that citizen science is of major importance as it supplies vital information for future scientific endeavors without the need to harm any animals, thus supplementing e-DNA as a non-invasive sampling approach for assessing community structure and biodiversity issues in marine habitats. However, its true potential to assist or even replace conventional sampling methods for planning future scientific surveys requires thorough further investigation.

Up to now, Croatia has established 289 Marine Protected Areas (MPA) that abide by European Union Natura 2000 guidelines (Figure 4) [80]. These MPAs cover an area of 5117 km^2^, which is 9.2% of Croatia’s Exclusive Economic Zone [80]. However, 239 of these zones have only been designated but have not been implemented yet [80]. Thus, only 2.2% of Croatia’s marine areas have actually implemented minor protection measures [80]. Croatia’s MPA administrations and practitioners are connected through the common network initiatives AdrionPAN and MedPAN that strive to improve conservation in these areas. A study published in Science in 2018 found, however, that 59% of Marine Protected Areas in the Mediterranean Sea are commercially trawled and that average trawling intensity is even 1.4-fold higher than in non-protected areas, thus also threatening vulnerable elasmobranch species [108]. Whether this is also the case in Croatia’s Exclusive Economic Zone requires further investigation. Strikingly, the distributions of chondrichthyans found in the current study often do not match the locations of Marine Protected Areas, reflecting the potential lack of conservation measures in designated but not properly implemented MPAs. This is especially the case for Rijeka in the Kvarner Bay, as well as for the Split area, where many species were found in the Brač channel (Figure 3). It may be possible that fishing is indeed reduced in Marine Protected Areas and unprotected areas thus supply more records. This may indicate particular biodiversity hotspots that need further protection, especially since many records from these regions were collected by beachgoers, scuba divers and snorkelers, showing that the animals lived at that location instead of only being landed there and caught somewhere else. There is an urgent need for independent monitoring of sharks, rays and skates in fishery statistics. Knowledge about the occurrences of certain rare species has the potential to provide vital information for planning future scientific surveys to identify, for example, possible nursery areas and aggregations of populations that are in need of immediate protection measures.

Three different categories of species can be inferred from our evaluation here. The first includes species that are extremely rare with an IUCN Red List status of critically endangered in the Mediterranean Sea, with records of only a few individuals, the last of which were reported a long time ago. These species most probably include *Leucoraja circularis*, *Pristis pectinata*, *Squatina aculeata*, *Squatina oculate*, and *Sphyrna zygaena*. The second category represents species that usually live in deeper waters, so that scuba divers, snorkelers and beachgoers do not normally encounter them. Last occurrences are also quite old and infrequent, but this does not necessarily mean that these species are rare per se. All other species found during this study currently have established populations in Croatia, many of which are declining. Some populations in the Mediterranean Sea are even declining very quickly considering their historical levels [8,12,73,109,110], but the actual extent needs further assessment.

Overall, commercial fishing poses by far the biggest threat to cartilaginous fishes in Croatian waters and the Mediterranean Sea [73,109,110]. Recreational fishing (see Figure 2D) also threatens endangered species, continually diminishing their diversity. A recent resolution prohibits recreational fishing for 39 species of sharks, rays, and skates in the Mediterranean Sea (GFCM/45/2022/Annex 1) [111].

In order to properly protect chondrichthyans in the Adriatic Sea, fishing activities must be thoroughly monitored and controlled, and existing regulations must be enforced. The impact of fishing must be considerably reduced through fisheries management measures, including carefully designed temporal and spatial protective measures as well as technical measures to reduce accidental bycatch, especially in areas where endangered species prevail. Citizen science can give important insights into these areas and provide a baseline for future research.

## Figures and Tables

**Figure 1 biology-12-00952-f001:**
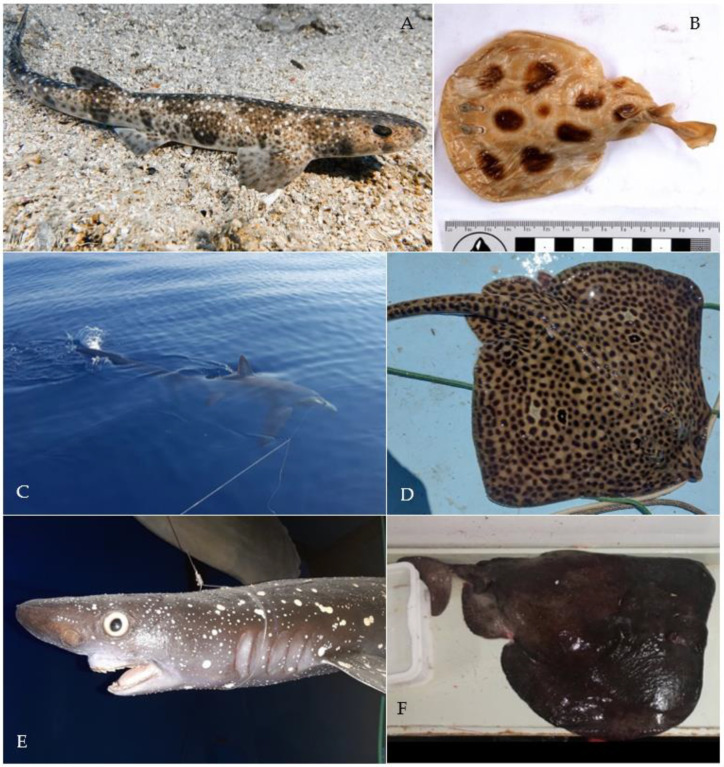
Photos of rare chondrichthyan species found in Croatian waters. (**A**) *Scyliorhinus duhamelii* (Garman, 1913) taken on 9 July 2021 in Selzine near Cres at 37 m depth by Matthias Brunner. Characteristic dark spots that form aggregations and white spots are visible. (**B**) Male juvenile *Torpedo torpedo* (Linnaeus, 1758) (inventory number NHMW—87406) caught near Split in 1916, preserved in the Natural History Museum Vienna. (**C**) *Alopias superciliosus* (Lowe, 1841) caught with a longline near Dubrovnik in 2017 by Maro Sekula. (**D**) *Raja polystigma* Regan, 1923 caught near the island Ist at 40 m depth on 19 July 2022, photo by Dario Marinov. (**E**) Male *Echinorhinus brucus* (Bonnaterre, 1788), preserved taxidermied in the Museum of Natural History of Venice, Fontego dei Turchi (inventory number 7781), caught in the Kvarner Gulf on 5 May 1877, Museum of Natural History of Venice, G. Ligabue, photographic archive. (**F**) *Tetronarce nobiliana* (Bonaparte, 1835), weighing 17.5 kg, caught on 17 April 2018 south of Slano by bottom trawling at 250 m depth by Dubravko Behlulović: Hrvatski koćari.

**Figure 2 biology-12-00952-f002:**
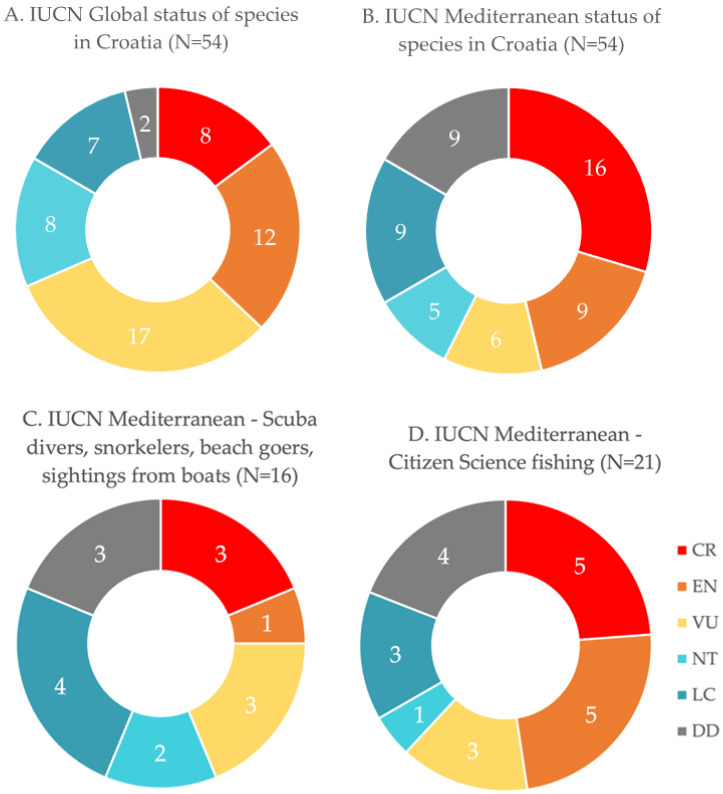
Percentages of IUCN statuses of records for (**A**) all species from Croatian waters with global IUCN status, (**B**) all species from Croatia with Mediterranean IUCN status, (**C**) species collected from citizen science by scuba divers, snorkelers, beachgoers, and sightings from boats with Mediterranean IUCN status, and (**D**) species collected from citizen science by fishers with angling, longlining, gill nets, beach seine and trawling, with Mediterranean IUCN status. Numbers show the number of species per IUCN category. Critically endangered (CR) = in a particularly and extremely critical state; Endangered (EN) = very high risk of extinction in the wild; Vulnerable (VU) = at high risk of unnatural (human-caused) extinction without further human intervention; Near threatened (NT) = close to being endangered in the near future; Least concern (LC) = unlikely to become endangered or extinct in the near future; Data deficient (DD) [12].

**Figure 3 biology-12-00952-f003:**
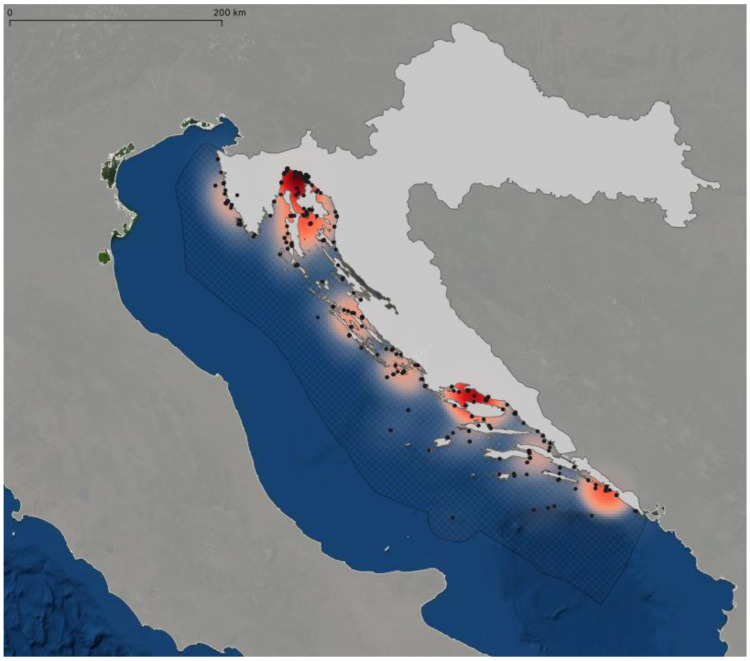
Heatmap showing the distribution and abundance of all species of Chondrichthyes in Croatia combined, based on data collected during this study. The grid shows the boundary of Croatia’s Exclusive Economic Zone (EEZ). Black dots indicate the location of individual records (see Appendix A for exact location). Note the prevalence of occurrences in the cities of Rijeka, Split and Dubrovnik.

**Figure 4 biology-12-00952-f004:**
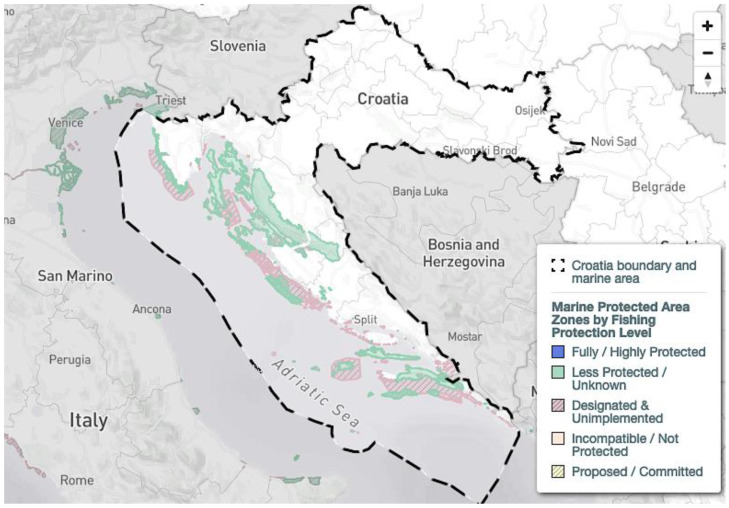
Marine protected areas (MPAs) in Croatia with the boundary of Croatia’s Exclusive Economic Zone (EEZ) in dashed lines. Map obtained from the Marine Protection Atlas, Marine Conservation Institute [79,80].

**Figure 5 biology-12-00952-f005:**
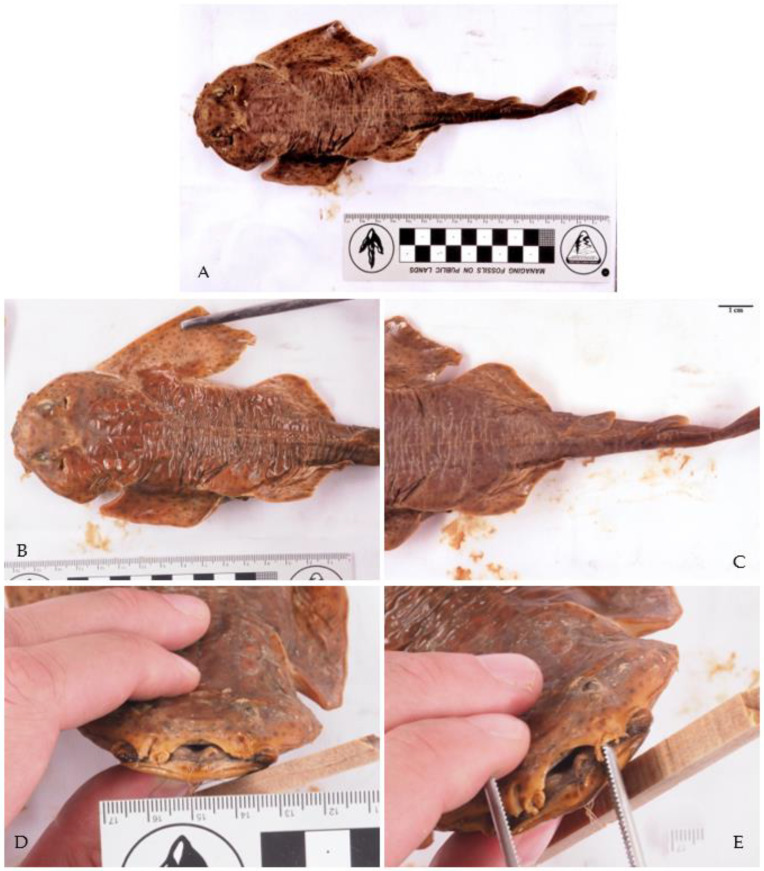
Specimen of *Squatina aculeata* Cuvier, 1829 from the Natural History Museum Vienna (inventory number NHMW—88925). (**A**) Whole specimen, scattered with small dark spots; large prominent thorns are present on its midback; no large ocelli are visible. (**B**) The width of the pectoral fin is half as long as its length. (**C**) First dorsal fin originates over the rear tips of the pelvic fin. (**D**,**E**) Nasal barbels and anterior nasal flaps are heavily fringed, with no straight or spatulate tip present, and the lateral head folds only have weak lobes.

**Figure 6 biology-12-00952-f006:**
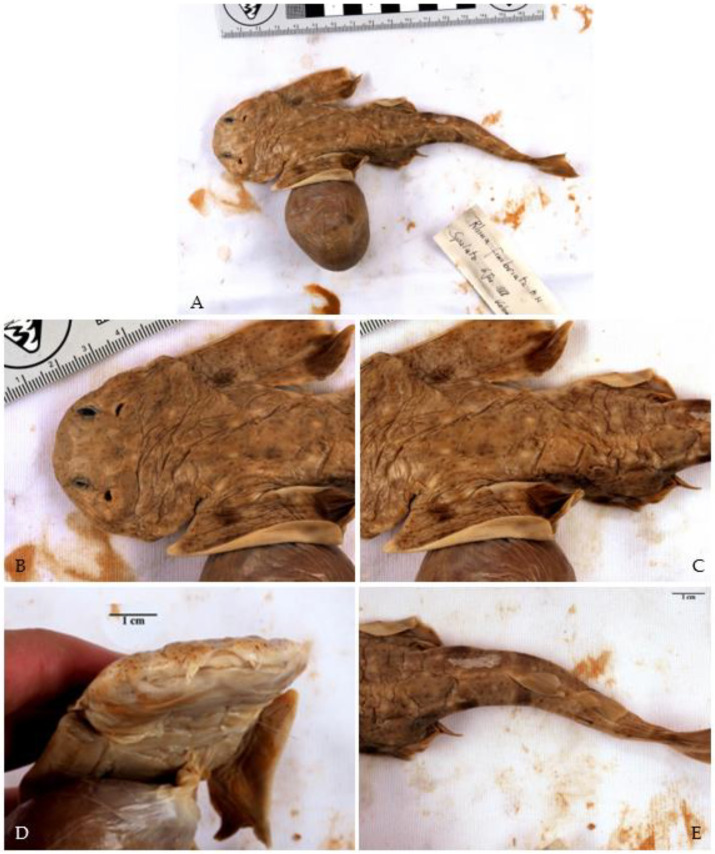
One of two specimens of *Squatina oculata* Bonaparte, 1840 from the Natural History Museum Vienna (inventory number NHMW—22429). (**A**) Complete specimen with large symmetrical dark blotches visible on the pectoral fins as well as on the tail, and the width of the pectoral fins is about half its length. (**B**) Head with no thorn-like placoid scales on the snout and between the eyes. (**C**) Midback line without thorns. (**D**) Head, nasal barbels and anterior nasal flaps, as well as a concave interorbital space, are clearly visible; the lateral head folds are without lobes. (**E**) First dorsal fin is well behind the posterior tips of the pelvic fin.

**Figure 7 biology-12-00952-f007:**
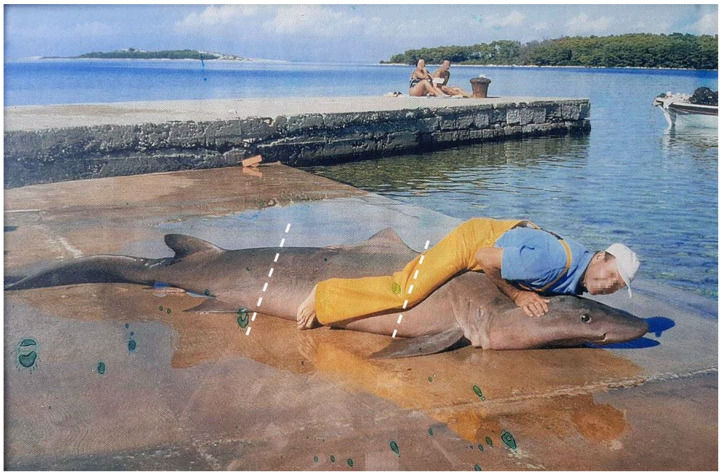
Photograph of an approximately 380 cm-long *Odontaspis ferox* (Risso, 1810) from the Island of Molat, captured in September 1999. Dashed lines indicate the origins of the first dorsal fin and the pelvic fin, which is an important feature to distinguish between *O. ferox* and *C. taurus*. Photograph by Krešimir Matulić and Baranić Ante.

**Figure 8 biology-12-00952-f008:**
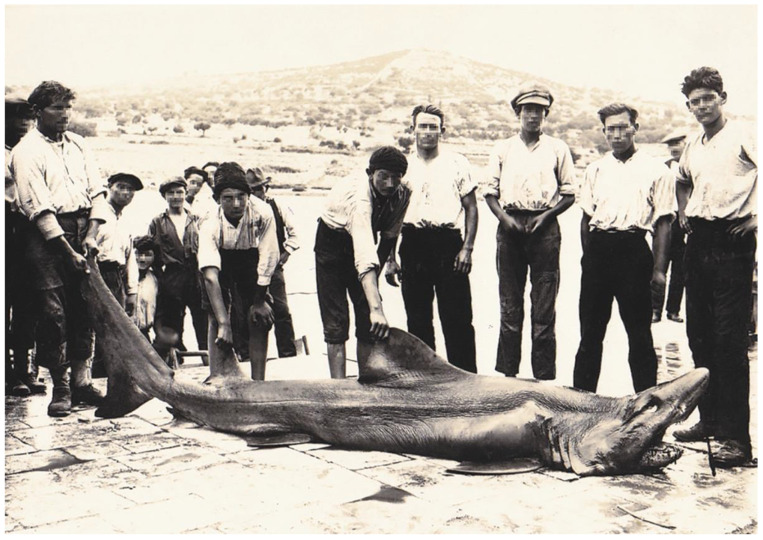
Photograph of a smalltooth sandtiger shark, *Odontaspis ferox* (Risso, 1810), from Stari Grad on the island of Hvar, captured in 1911. Note the position and size of the first dorsal fin. Copyright: Igor Goleš, 2022 © Forgotten Dalmatia [50].

**Table 1 biology-12-00952-t001:** Updated checklist of chondrichthyan species recorded in Croatian waters, with first and last records, historical data (Hist), literature data (Lit), citizen science data (Cit), the IUCN statuses for Global and Mediterranean areas, and references. Abbreviations for references used: MECO, Mediterranean Elasmobranch Citizen Observations; WWF, World Wide Fund for Nature Adria; NHMW, Naturhistorisches Museum Wien; PMS, Prirodoslovni Muzej Split; PMR, Prirodoslovni Muzej Rijeka; PMD, Prirodoslovni Muzej Dubrovnik; UNIFI, Museo di Storia Naturale dell’Università di Firenze. Asterisks used: * species newly recorded here; ** presence of species unconfirmed. For more information, see extended discussion.

Taxon	First Record	Last Record	Hist	Lit	Cit	IUCN Global/Med	References
**Order CHIMAERIFORMES**							
**Family Chimaeridae Rafinesque, 1815**							
*Chimaera monstrosa* Linnaeus, 1758	1962	24 June 1996	2	1		VU/NT	[29], PMS, PMR
**Order RAJIFORMES**							
**Family Rajidae (de Blainville, 1816)**							
*Dipturus* cf. *batis* (Linnaeus, 1758)	28 November 1879	1970	2	3		CR/-	[24,30,31], PMD, UNIFI
*Dipturus nidarosiensis* (Storm, 1881)	21 August 2008	12 May 2010		1		NT/-	[32]
*Dipturus oxyrinchus* (Linnaeus, 1758)	2 June 1950	17 June 2010	1	34		NT/NT	[24,29,33], PMR
*Leucoraja circularis* (Couch, 1838)	15 November 1950	13 November 1952		8		EN/CR	[24]
*Leucoraja fullonica* (Linnaeus, 1758)	-	-	-	-	-	VU/CR	[34,35]
*Raja asterias* Delaroche, 1809	1848	1994–1998	11	1		NT/NT	[30,33], PMD, NHMW, PMS
*Raja clavata* Linnaeus, 1758	1842	19 August 2021	10	98	7	NT/NT	[24,29,30,31,33,36], PMS, PMR, PMD, NHMW, UNIFI, MECO, WWF
*Raja miraletus* Linnaeus, 1758	1882	19 May 2021	5	5	1	LC/LC	[24,29,30,33,36], PMS, PMD, PMR, NHMW, WWF
*Raja montagui* Fowler, 1910	1883	27 February 2003	3	6		LC/LC	[24,29,33,37], PMS, PMR, NHMW
*Raja polystigma* Regan, 1923	1987	19 July 2022	2		1	LC/LC	[30], PMD, NHMW, WWF
*Raja radula* Delaroche, 1809		1984		1		EN/EN	[29,38]
*** Raja undulata* Lacepède, 1802	-	-	-	-	-	EN/NT	[39]
*Rostroraja alba* (Lacepède, 1803)	1 October 1879	30 November 2022	6		2	EN/EN	[30,31], PMD, UNIFI, NHMW, PMR, WWF
**Order MYLIOBATIFORMES**							
**Family Dasyatidae Jordan and Gilbert, 1879**							
*Bathytoshia lata* (Garman, 1880)	1953	29 August 2002		3		VU/-	[33,40,41]
*Dasyatis pastinaca* (Linnaeus, 1758)	1881	27 September 2021	2	1	10	VU/VU	[30,33], PMD, NHMW, MECO, WWF
*Pteroplatytrygon violacea* (Bonaparte, 1832)	19 July 2007	15 August 2021	4		10	LC/LC	PMR, WWF, MECO
**Family Gymnuridae (Fowler, 1934)**							
*Gymnura altavela* (Linnaeus, 1758)	1927	5 August 2022	2	1	1	EN/CR	[30,40] PMS, PMD, WWF
**Family Myliobatidae Bonaparte, 1835**							
*Aetomylaeus bovinus* (Geoffroy Saint-Hilaire, 1817)	before 1989	23 December 2020	1		4	CR/CR	[30], PMD, MECO
*Myliobatis aquila* (Linnaeus, 1758)	1882	27 July 2021	3	1	3	EN/VU	[30,33], PMD, PMR, WWF, MECO, NHMW
**Family Mobulidae Gill, 1893**							
*Mobula mobular* (Bonnaterre, 1788)	20 September 2019	8 September 2021			4	CR/-	MECO, WWF
**Order TORPEDINIFORMES**							
**Family Torpedinidae Henle, 1834**							
*Tetronarce nobiliana* (Bonaparte, 1835)		17 April 2018			1	LC/-	MECO, Dubravko Behlulović
*Torpedo marmorata* Risso, 1810	1882	3 July 2021	9	1	14	VU/LC	[30,33], PMS, PMD, PMR, WWF, MECO, NHMW
*Torpedo torpedo* (Linnaeus, 1758)	1916	1994–1998	1	1		VU/LC	[33], NHMW
**Order RHINOPRISTIFORMES**							
**Family Pristidae Bonaparte, 1835**							
*Pristis pectinata* Latham, 1794	1901	1929	2	3		CR/CR	[37,42], PMS, PMD
**Family Rhinobatidae Bonaparte, 1835**							
*** Rhinobatos rhinobatos* (Linnaeus, 1758)	-	-	-	-	-	EN/EN	[17]
**Order HEXANCHIFORMES**							
**Family Hexanchidae Gray, 1851**							
*Heptranchias perlo* (Bonnaterre, 1788)	25 June 2020	3 March 2022		2		NT/DD	[43]
*Hexanchus griseus* (Bonnaterre, 1788)	18 April 1905	4 September 2021	1	5	4	NT/DD	[24,44], PMR, WWF, MECO
**Order ECHINORHINIFORMES**							
**Family Echinorhinidae Gill, 1862**							
*Echinorhinus brucus* (Bonnaterre, 1788)	5 May 1877	5 May 1877		1		EN/EN	[45,46]
**Order SQUALIFORMES**							
**Family Dalatiidae Gray, 1851**							
*Dalatias licha* (Bonnaterre, 1788)		1984		1		VU/EN	[29]
**Family Squalidae de Blainville, 1816**							
*Squalus acanthias* Linnaeus, 1758	1840	26 July 2021	20	2	2	VU/EN	[29,30,31,33], PMD, PMS, PMR, UNIFI, NHMW, WWF
*Squalus blainville* (Risso, 1827)	1829	1994–1998	4	2		DD/DD	[29,30,33], PMD, NHMW
**Family Centrophoridae Bleeker, 1859**							
*Centrophorus* cf. *uyato* (Rafinesque, 1810)	12 July 1950	1984		46		EN/CR	[24,29]
**Family Etmopteridae Fowler, 1934**							
*Etmopterus spinax* (Linnaeus, 1758)	26 April 1951	1984	2	17		VU/LC	[24,29,30], PMD, PMR
**Family Oxynotidae Gill, 1863**							
*Oxynotus centrina* (Linnaeus, 1758)	1927	1 September 2021	6	21	3	EN/CR	[29,30,37,47,48], PMD, PMS, WWF, PMR
**Order SQUATINIFORMES**							
**Family Squatinidae de Blainville, 1816**							
** Squatina aculeata* Cuvier, 1829		April 1939	1	1		CR/CR	[49], NHMW
*Squatina oculata* Bonaparte, 1840	6 January 1888	21 July 1893	1	1		CR/CR	[49], NHMW
*Squatina squatina* (Linnaeus, 1758)	1883	4 June 2021	14	5	2	CR/CR	[29,30,37,49], NHMW, WWF, PMS, PMD, PMR
**Order LAMNIFORMES**							
**Family Odontaspididae Müller and Henle, 1839**							
*Odontaspis ferox* (Risso, 1810)	1911	1954	2	1		VU/CR	[30,50,51,52,53], PMD
**Family Carchariidae Müller and Henle, 1839**							
*** Carcharias taurus* Rafinesque, 1810	-	-	-	-	-	CR/CR	[52,53,54,55,56,57,58]
**Family Alopiidae Bonaparte, 1835**							
** Alopias superciliosus* (Lowe, 1841)		2017			1	VU/EN	MECO, Maro Sekula
*Alopias vulpinus* (Bonnaterre, 1788)	1890	8 July 2021	1		3	VU/EN	[30], PMD, MECO, WWF
**Family Cetorhinidae Gill, 1861**							
*Cetorhinus maximus* (Gunnerus, 1765)	1822	30 March 2021	1	30	3	EN/EN	[57,59,60,61,62,63], PMR, MECO
**Family Lamnidae Bonaparte, 1835**							
*Carcharodon carcharias* (Linnaeus, 1758)	14 September 1868	6 October 2008	2	94		VU/CR	[30,51,57,62,64,65,66,67,68,69], PMD, PMR
*Isurus oxyrinchus* Rafinesque, 1810	1871	8 June 2019	2	45	1	EN/CR	[30,37,57,70,71], PMD, MECO
*Lamna nasus* (Bonnaterre, 1788)	19 September 2018	19 September 2018		1		VU/CR	[72]
**Order CARCHARHINIFORMES**							
**Family Pentanchidae Smith, 1912**							
*Galeus melastomus* Rafinesque, 1810	1948	24 June 1996	5	108		LC/LC	[24,29,30,37], PMR, PMD, PMS
**Family Scyliorhinidae Gill, 1862**							
*Scyliorhinus canicula* (Linnaeus, 1758)	27 October 1879	13 June 2021	10	49	3	LC/LC	[24,29,30,31,33], PMD, PMS, PMR, UNIFI, MECO
**/** Scyliorhinus duhamelii* (Garman, 1913)	28 June 2021	4 September 2021			4	-/-	WWF
*Scyliorhinus stellaris* (Linnaeus, 1758)	1 October 1879	16 June 2021	10	2	8	VU/NT	[29,30,31,33], PMR, PMS, PMD, UNIFI, WWF, MECO
**Family Triakidae Gray, 1851**							
*Galeorhinus galeus* (Linnaeus, 1758)	10 June 1950	20 December 1992	5	50		CR/VU	[24,30,37], PMD, PMR, PMS, NHMW
*Mustelus asterias* Cloquet, 1819	1948–1949	2008	1	6		NT/VU	[30,33,73,74,75], PMD
*Mustelus mustelus* (Linnaeus, 1758)	9 October 1879	20 August 2021	7	22	6	EN/VU	[24,30,31,33,73,74,75,76], PMD, PMS, PMR, UNIFI, NHMW, WWF, MECO
*Mustelus punctulatus* Risso, 1827	1930–1939	25 July 2021		8	3	VU/VU	[73,74,75], WWF
**Family Carcharhinidae Jordan and Evermann, 1896**							
*Carcharhinus plumbeus* (Nardo, 1827)	1895	30 July 2021	5	2	2	EN/EN	[24,30,37,77], PMD, PMR, PMS, NHMW, MECO, WWF
*Prionace glauca* (Linnaeus, 1758)	9 August 1883	10 July 2021	7	13	11	NT/CR	[24,30,37,57,78], PMD, PMS, WWF, MECO
**Family Sphyrnidae Bonaparte, 1840**							
*Sphyrna zygaena* (Linnaeus, 1758)	1873	1 November 1956	3			VU/CR	[30], PMS, PMD, PMR
*** Sphyrna tudes* (Valenciennes, 1822)	-	-	-	-	-	CR/-	[39]

## Data Availability

All data used by the authors for the analysis are available in the Appendix A.

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
