# Peer review of "Updated Checklist of Chondrichthyan Species in Croatia (Central Mediterranean Sea)"

_biology, 2023, doi:10.3390/biology12070952_

Round 1

Reviewer 1 Report

Personally, I do not like the use of 'Citizen science record' as those records are not aimed at to be of 'scientific' use, although we can use them.  I prefer to use 'people records' or 'casual records' as they are from several different categories (fishermen, tourists, snorkelers....)

Related to the above records, I guess that the problems related (mainly geographical concentration and environmental biases) must be better underlined in the conclusions.

Author Response

We want to thank reviewer 1 for their positive assessment of the manuscript and comments. We agree that the term “citizen” does not accurately describe the different categories of people who provided data, and “scientific” can be misleading since the people who collected the data are not scientists themselves. However, the term “citizen science” is a generic term that is used to describe data that was collected by the general public who contribute to science. Since this term is commonly used in scientific articles, we think that it is the right one for our kind of data.

Regarding your second point we fully agree and included a more detailed description regarding the problems and biases of citizen science in our conclusion.

Reviewer 2 Report

I enjoyed the reading the manuscript. The fish group, condrichthys is less studied but vulnerable taxa. The authors have updated the list from Adriatic Sea. I have some minor suggestions:

1. new records and new species are not same. When choosing these terminology please be careful. New records are described earlier from other parts of the world but not the new species. Please check throughout the MS.

2. Please validate your list comparing with the established database, WoRMs.

3. Please correct using present and past tenses even in the Abstract.

 Thank you 

Needs English check. 

Author Response

We want to thank reviewer 2 for their affirmative assessment and for pointing out these inconsistencies. We checked the terminology throughout the manuscript and changed it accordingly. Of course, we validated our list with WoRMS, although it was not mentioned in the manuscript. Please find this additional validation now included in the methods as well. Also, our English native speaking colleague Dr. John Joseph Cawley kindly took the time to revise the English in our manuscript. We hope these improvements are satisfactory.

Reviewer 3 Report

The manuscript provides a good review of the Chondrichthyan's diversity in the Adriatic Sea. English should be reviewed because is not easy in some parts to understand, redaction should be improved. In methods I recommend specifying if you make the search in several languages or only in English because this area is surrounded by countries in which the principal language is not English I recommend to use several languages in the search.  Also, you do not explain what was the criteria to exclude or reject publications for your search. In results, you use the term sighting but they are not sighting are records or reports. Sighting should be used only for Citizen Science data. I recommend to add in the table which of the reports are confirmed and which are not in this way the readers can see easily the species that are not confirmed for the region, also in this section, you mix discussion. In the discussion, you only repeat the results that you present in the table, I consider it is too long and lacks a proper discussion on the data comparing with previous reports,  for example: ¿why you found more species?, ¿why is different from the Mediterranean?. The conclusion is too long also. 

English should be improved, they use some words that are too hard to understand and also the redaction should be improved. 

Author Response

We want to thank the reviewer for her positive feedback and highly appreciate the thoroughness with which she read the manuscript, which proves her dedication to help us improve it. To improve the English, we handed our manuscript to a native English speaker, our colleague Dr. John Joseph Cawley, and included all his changes.

Thank you for being so attentive and for pointing out the search in other languages. We indeed forgot to mention the use of Croatian key words in our manuscript, although we of course did use them. Two of our co-authors, Pero Ugarković and Patrik Krstinić are Croatian citizens themselves and contributed Croatian literature to our search.

Thank you for pointing out the excluded publications we mentioned in the methods. Of course, we did not exclude or reject any publications, and included everything we could find. This half sentence must be a remnant of the work process during the writing of the bachelor’s thesis of the first author, which this manuscript originally was. We now deleted this half sentence in the manuscript to prevent any confusion.

Thank you for pointing out the inconsistent use of the term “sightings”. We agree that it should only be used for citizen science data. We now changed the terminology throughout the manuscript to make it more consistent.

In Table 1 we included the marking “** presence of species unconfirmed”, which is positioned before the species name in the table. We hope that this highlights the unconfirmed and doubtful species for the reader at first sight.

We found more species than the last three checklists due to new data we acquired through citizen science and the Natural History Museum of Vienna. We understand that the section comparing our list with the previous checklists has a discussing character. However, since we primarily highlight the additional and missing species in our list compared to the previous works, and our work is an updated checklist, we think that this summary is important to be placed in the results. We then provide further information for every species in our checklist in the discussion, not only presenting the results but adding exhaustive information and context which might be of interest to the reader. This includes the current IUCN status, information on their Mediterranean locality, the currently known distribution in the context of the biology of the respective species, common identification problems which are important to know when considering incomplete fishery data, further explain historical as well as new records, and the protection status in Croatia. We do not think the conclusion is too long, as it summarizes the context of our data collection methods and contributes some additional information to conservation efforts for cartilaginous fishes in Croatia, without any redundant information.

We did our best to incorporate all the improvements by reviewer 3 and hope this version of the manuscript is now satisfactory.

Reviewer 4 Report

Comments on the manuscript:

Comment 1

In the Introduction you mention that the Croatian Ministry and the EU do not publish much information on the elasmobranchs status. However, there are annual reports to have a look at from the DCF program, between 2013 and 2021, for Croatia and since 2004 for many other countries.

Comment 2

In the results you are mentioning some species that have been reported occurring in the Adriatic Sea, and some in the Croatian waters, but you have not observed them in your search or the citizen science reports. What about the Data Collection Framework of your country? Do they mention these species in their reports, from onboard records or the MEDITS surveys? Are you sure they need further investigation or these have been reported in annual reports from the Ministry?

Comment 3

How did you identify Scyliorhinus duhamelii? You are presenting a photograph of a dogfish inside the sea, without clear evidence of the species. Did you had the chance to measure it, or carried out DNA to confirm its occurrence? I am not sure of its existence in the Mediterranean Sea, I have looked for papers confirming its occurrence based on genetic analysis mostly, but could not find any. Please state how did you identify the species, since identifications through pictures have caused many problems lately in the basin, and it is recommended to be avoided. Therefore, you might have to consider this as a valid species in the area!

Comment 4

All figures and tables need to follow the text. It is not easy to look for them in 31 pages to understand the text. Thus, right after you mention, for example Table 1, the table should be positioned right below the paragraph.

Comment 5

I would recommend having the document corrected by an English speaking researcher, since there are many mistakes in the text highlighted to be rephrased or changed. And also to have a look at the records from the Ministry of Croatia published in technical reports, to confirm species occurrence that might be doubtful to you or have not been observed by your team and citizen science observations.

Comment 6

Why do you propose to apply measures without knowing the status of populations and species? Rare species might be rare due to their bathymetric distribution (>200m) at which fisheries is not applied regularly (or at all), and thus their catches are considered rare. Threatened species is another case but we cannot put in the same basket all chondrichthyans! Be careful of what you present in the Discussion in order to avoid any confusion of the readers.

English grammar needs some improvement and authors should be careful in some sentences as they are confusing. I would suggest to give the manuscript to an English speaking researcher to make the appropriate corrections.

Author Response

We highly appreciate the thoroughness of this reviewer who clearly took the time to carefully read the manuscript in order to contribute essential improvements.

(1,2) Thank you for pointing out the DFC program, which we indeed forgot to mention and will now include in the introduction. The DFC from Croatia indeed has some data listed on specific species. However, only 23 chondrichthyan species are mentioned with minimal information on catches, all of which are already included in our list. Therefore, no changes regarding our list are required, but it is still very important to mention it in our manuscript to make it more complete.

(3) Scyliorhinus duhamelii is a valid species according to WoRMS and FishBase and is known to occur in the Mediterranean Sea (see Soares and Carvalho 2019 as well as Soares and Carvalho 2020). Ebert and Dando 2020 even proposed S. duhamelii to be endemic to the Mediterranean. We identified S. duhamelii based on the coloration from photographs, described in the morphological diagnosis provided by Soares and Carvalho 2019:  “Scyliorhinus duhamelii differs from all congeners by presenting a color pattern composed of scattered dark spots of varied sizes that also form aggregations ([…] vs. dark spots predominantly greater than spiracles in S. stellaris […], spots with well defined borders and not forming aggregations in S. canicula […]).“ Associated with these, we also observed prominent white spots on all animals. The drawing in Ebert and Dando 2020 well depicts the color pattern of dark spots that form aggregations as well as white spots, which we saw in every photo we had of this species. We therefore decided to use a different picture for the manuscript, by the same photographer, Matthias Brunner, that better shows these patterns and therefore hope to improve the understanding of our reasoning in the manuscript.

Since we only had data from citizen science in the form of photographs from divers or from the fishmarket, we were restricted to a mere morphological identification of this species. We are aware that this is not ideal and therefore describe and discuss these problems in our discussion. We tried to address the concerns by the reviewer and made some changes to the discussion to describe the situation more precisely. In our paper we primarily want to initiate a dialog regarding this species, since it was never mentioned before in checklists from this region. Although S. duhamelii is a valid species according to WoRMS and FishBase, for the region we are discussing there are several possibilities that further studies need to address: (1) S. duhamelii might be a morphologically and genetically distinct species from S. canicula, therefore three catshark species may exist in Croatia; (2) S. duhamelii might be genetically the same as S. canicula and therefore just a color morph; (3) S. duhamelii might be a distinct species, but hybridizes with S. canicula. Since there is also the possibility, as proposed by Ebert and Dando 2020, that S. duhamelii might be an endemic species to the Mediterranean Sea, we think it is very important to highlight these possibilities. These might have an impact for our understanding of speciation in the Adriatic Sea, especially in the current situation where large top predators are endangered by fishing, and their position in the food chain might be replaced by smaller mesopredators such as catsharks. But, of course, we can not say for sure that this species exists in Croatia, we therefore described it as “** presence of species unconfirmed” in Table 1. Although we of course would like to, but it is not in the scope of this work to provide a clear answer to this issue. Rather we want to start a discussion and inspire future works. Therefore, molecular analyses were unfortunately not in the scope of this paper, but the first author plans to conduct these in her master’s thesis.

(4) Thank you for pointing out this problem. We accordingly rearranged some figures, so figures and tables strictly follow their first mentioning in the text.

(5) Our English native speaking colleague Dr. John Joseph Cawley kindly took the time to revise the English in our manuscript.

(6) We did not propose to apply any measures, but we can see that some sentences in the manuscript could seem to imply an intention into that direction. We deleted these sentences accordingly. The rarity of reports of bathyal species are of course mentioned and discussed for each species separately.

Round 2

Reviewer 3 Report

I appreciate that you follow some of the minor comments I made about your manuscript. As I can see and due to your response to the comments about your discussion and conclusion section, this manuscript seems to aim to do a review of the species distributed in Croatia and a comments list because the discussion format is more for a review paper than for a original scientific paper. What I expect for an original scientific paper is to discuss their findings and results and oppose explanation about what they found but in your manuscript the discussion only have a resume of the available information about the species you found reported in Croatia. This information is very interesting but is not new. So I recommend you reconsider the format of your manuscript and probably change it to a review paper. 

Author Response

We want to thank Reviewer 3 for her scrutiny and input. We agree that checklists usually resemble reviews as they compile available information on species in a certain area. However, we collected a range of additional data from citizen science and specimens from museums that we present and compare to previous findings. Here, we also want to bring your attention to the supplementary material we provided, as all new records and data we collected are extensively listed there. We also point out a range of species that were doubtful in the past, where we include a new outline on their status in the discussion, as is the case for Odontaspis ferox for example. Our work is therefore original research, and we therefore do think that our manuscript is an article.

In order to improve the manuscript according to the suggestions by the reviewer, we changed the discussion so that it discusses our new findings rather than giving the impression of a review. We moved the comparison to other checklists and the description of distribution patterns from the results to the discussion section, as these paragraphs have a discussing character, as the reviewer correctly pointed out. We then list the extended discussion on the respective species afterwards. We hope that these measures satisfy the reviewer and improves the allocation of discussing text in the manuscript.

Reviewer 4 Report

Dear authors,

Thank you for considering all the suggested corrections, however there are still some parts that need to be looked over again. I am attaching the manuscript with some comments and grammar corrections or expressions used, to take into account, prior to the Editor's decision.

In case you have used information on species from the Medits surveys, please mention this to the text where appropriate.

I would suggest to find someone from the field to look over the English grammar of the manuscript because I have found several mistakes in the expression in the text again.

Author Response

We want to thank Reviewer 4 again for their thorough investigation of the manuscript and suggested corrections. We appreciate all suggestions made by Reviewer 4 for the improvement of the manuscript and included them where we could. We want to sincerely apologize for the discontent of the English language. Our colleague Dr. John Joseph Cawley, who is an Irish native and an ichthyologist, had a close look at the English language of the manuscript. We are not sure why these corrections are not satisfactory, but it may be due to Dr. Cawley’s British English, in contrast to the American English that is more commonly used in scientific articles. Therefore, although the English is correct in spelling and grammar, it may be unfamiliar to some readers, for which we want to apologize. We included stylistic suggestions by Reviewer 4 and hope that this improves the readability of the manuscript.

Regarding the information on species from the Medits surveys we cited it whenever applicable. Please also see the Supplementary Table for detailed credits for every Medits record.

“It might be a valid species in databases, but it is not valid in the Mediterranean yet, as there are no published records to support this. You can mention that you believe it exists in the area but gets confused with S.canicula, however, evidence of morphometrics or genetics are not presented here (or anywhere else).

Ebert and Dando might have carried out genetics to support the species existence in the Med, but they have not published their research yet. The species is only mentioned in this book!
In addition, this book does not include the presence of Dasyatis marmorata in the Med, which has been recorded in 4 countries, with DNA sequencing.

My point is that we need to be very careful when including something new in the list of species occurring in the Med Sea; and opinions might be contrasting but need to be heard. I will be awaiting for your next work with evidence on the species occurrence!“

We apologize for the ongoing confusion regarding Scyliorhinus duhamelii. The lectotype of S. duhamelii is from the Adriatic Sea, and other specimens described in Soares and Carvalho 2019 are from the Mediterranean Sea and Croatia too. We therefore do not base the information on the locality of this species from Ebert and Dando 2020, but from Soares and Carvalho 2019 and 2020. We now included this information in the manuscript and hope that this describes the situation more precisely.

„This is a very BOLD statement. Any alternative method to replace or even supplement conventional sampling, requires validation prior to be accepted by the scientific community.

So far the only non-invasive technique for assessing community structure or biodiversity issues is e-DNA approach (used broadly in many countries lately).

To my knowledge citizen science lacks validation with concurrent experiments testing it against conventional sampling estimates. If so, please provide any relevant study validating citizen science against conventional sampling.

You may write about how important citizen science is at your work for collecting information on areas that research surveys do not take place, but you cannot suggest that the EU should ban the trawling surveys.

This sentence has to be removed.“

„Once again I find this very bold to suggest, as citizen science cannot replace true science.

Unless, you mean that they will assist you to plan future surveys to collect data on elasmobranchs. If that is the case then you need to correct this phrase as i mentioned above (i.e. assist to plan future surveys to collect information on these species).

However, biomass, distribution and abundance data is collected only through research surveys like the Medits project (bigger projects), and thus you should be careful of what surveys you wish to carry out from citizen science only.“

We agree that some statements in our conclusions require more backup by research. We want to thank Reviewer 4 for pointing them out, and we accordingly deleted these sentences or included additional references. We are very grateful for the interesting insights the reviewer provided on citizen science as a sampling method. We hope it is in accordance to their wishes to include some of that information in the text, as we find it very intriguing. Again, we want to thank Reviewer 4 for their thorough investigation and hope that our corrections are satisfactory.

Round 3

Reviewer 3 Report

This manuscript has improved a lot and in its present form has valuable information and provides a clear understanding of the evolution of the knowledge of chondrichthyans diversity in Croatia.